# Green and Efficient Acquirement of Unsaturated Ether from Direct and Selective Hydrogenation Coupling Unsaturated Aldehyde with Alcohol by Bi-Functional Al-Ni-P Heterogeneous Catalysts

**Yan Xu [†], Huiqing Zeng [†], Dan Zhao * [ID], Shuhua Wang, Shunmin Ding and Chao Chen ***

Key Laboratory of Jiangxi Province for Environment and Energy Catalysis, School of Chemistry and Chemical Engineering, Nanchang University, Nanchang 330031, China

* Correspondence: zhaodan@ncu.edu.cn (D.Z.); chaochen@ncu.edu.cn (C.C.); Tel.: +86-15879176996 (D.Z.); +86-15179167359 (C.C.)

† These authors contributed equally to this work.

**Abstract:** In view of the industrial importance of high-grade unsaturated ether (UE) and the inconvenience of acquiring the compound, herein, a series of low-cost Al-Ni-P catalysts in robust $AlPO_4/Ni_2P$ structure possessing novel bi-functional catalytic features (hydrogenation activation and acid catalysis) were innovated, and testified to be efficient for directly synthesizing UE with a superior yield up to 97% from the selective hydrogenation coupling carbonyl of unsaturated aldehyde (cinnamaldehyde or citral) with C1–C5 primary or secondary alcohol under 0.1 MPa $H_2$ and 393 K. The integrated advantages of high efficiency, green manner and convenient operation of the present heterogeneous catalytic system gave the system potential for feasibly harvesting high-grade unsaturated ether in related fine chemical synthesis networks.

**Keywords:** unsaturated ether; selective hydrogenation coupling reaction; ambient pressure hydrogenation; Al-Ni-P composites; bi-functional catalytic interface

## 1. Introduction

Unsaturated ether (UE) is a kind of high-grade ether that can be widely employed as a solvent, perfume and medical anesthetics [1–3]. The compound could also serve as an important intermediate for pharmaceutical preparation such as the synthesis of zampanolide, furaquinocin E and dihydro-β-erythroidine [4–6], since C-O-C is the feature fragment of these drug molecules, while the reserved C=C bond in the molecule chain is of great importance for facilitating further branch design or desirable functional bond addition. However, the acquirement of unsaturated ether has long been an arduous task in the network of fine chemical synthesis. The traditional Williamson method was only available for the synthesis of normal ether, which required the utilization of a series of hazardous additives [7,8]; the dehydrogenation of saturated ether was confined by the functional group and chain structure of the substrate, while the C-O-C bond could be easily broken in the dehydrogenation process, which would undergo a low selectivity to objective UE product [9,10]; in addition, homogeneous catalysts, including some environmentally unfriendly reagents such as liquid acid or base, were always employed in these reactions systems, which not only greatly lowered the greenness and environmental friendliness of the system, but also relied on a complicated post-processing for the separation and purification of the product, further impeding the application potential of these reaction systems.

Instead of the above methods, in 2008, Milone et al. first found that cinnamyl ethyl ether was present in the hydrogenation of cinnamaldehyde in ethanol over $Au/TiO_2$ with a considerable yield of 17% [11]. In 2020, Nasiruzzaman Shaikh et al. made progress by obtaining cinnamyl methyl ether with a yield of up to 97% from the hydrogenation of

cinnamaldehyde in methanol over Pt@GS under 3.0 MPa $H_2$, but the yield of UE could be radically suppressed below 15% when secondary alcohol was used [12]. To the best of our knowledge, these two reports are the only references to show the possibility that unsaturated ether (UE) could be synthesized from the hydrogenation coupling of unsaturated aldehyde (UA) with alcohol, the more convenient and greener route for directly acquiring UE than popular methods. Nevertheless, these findings have been neither widely noted nor further developed as a strategy to acquire UE, which could be attributed to the following formidable barriers: (1) the low catalytic efficiency to yield unsaturated ether even using precious metal catalysts; (2) the low selectivity to hydrogenation coupling reaction between carbonyl and alcohol when a C=C bond exists; (3) the confined substrate scope for both unsaturated aldehyde and alcohol. In addition, the harsh reaction conditions from the high pressure of $H_2$ required for achieving an acceptable yield of UE is another disadvantage of the route.

Confronting these barriers, herein, we formulated a series of heterogeneous Al-Ni-P catalysts for the hydrogenation transformation of cinnamaldehyde or citral in C1-C5 alcohol. Cinnamaldehyde or citral were chosen as typical reaction substrates due to the fact that the α, β-unsaturated carbonyl compound was the primary chemical which could be readily obtained from the large-scale Aldol condensation industry or a sustainable bio-refinery with biomass feedstock, and the selective hydrogenation transformation of such a compound was a crucial and challenging step in many fine chemical synthesis networks [13,14]. Aside from unsaturated aldehyde, the potential of reaction was also impacted by the feature of alcohol [11,12]. Therefore, a scope of alcohol was further employed in this investigation. It was demonstrated that a scope of unsaturated ether could be successfully synthesized from the selective hydrogenation coupling of carbonyl with alcohol to achieve an attractive yield over the cost-competitive and robust Al-Ni-P catalysts under mild reaction conditions, in particular under 0.1 MPa $H_2$, suggesting that the system could be referred to as an applicable technique to acquire UE in a highly efficient green and convenient manner for the advanced synthesis of corresponding fine chemicals.

## 2. Results and Discussion

### 2.1. Structure Feature of Al-Ni-P Catalysts

Al-Ni-P samples with an Al:Ni molar ratio of 0.05, 0.1 and 0.2 testified using the ICP-OES technique were derived from an easily handled one-pot solvent thermal reaction system, in which $HPO_4^{2-}$ was employed as the P-donor. The crystalline and surface feature of Al-Ni-P composites were firstly measured using XRD and XPS techniques, and the spectra are given in Figure 1. As shown, with the $AlPO_4$ diffraction (PDF cards No.#73-1977) for Al-P and $Ni_2P$ diffraction (PDF cards No.#74-1385) for Ni-P as reference, all three Al-Ni-P samples are present as $AlPO_4/Ni_2P$ crystalline mixture; accompanying the observation, the simultaneous occurrences of $Ni^{\delta+}$ and $P^{\delta-}$ photo-emissions resolved from Ni 2p (ca. 853.2 eV) and P 2p (ca. 130.0 eV) XPS spectra over Ni-containing samples further clarify $Ni_2P$ existence on the surface of these samples [15,16]. Meanwhile, $AlPO_4$ surface distributions among Al-containing samples are reflected by Al(III) photo-emission observed around 75.4 eV in the Al 2p spectra and P(V) board peaks located in the range of 132–136 eV in the P 2p spectra [17,18]. Moreover, as disclosed in Table 1, the surface Al:Ni ratio is clearly higher than the total Al:Ni ratio among Al-Ni-P composites, suggesting that $AlPO_4$ could enrich the surface of these samples. This suggestion was further proven through the EDS mapping from HRTEM measurements; as shown in Figure 2, the feature d-space of 0.223 nm belongs to $Ni_2P$ [11] crystalline (PDF cards No.#74-1385) exhibits on the dark particles sized around 200 nm for Ni-containing samples, while Al element distributions are shown as the small scatters surrounding these $Ni_2P$ particles for Al-Ni-P samples. Linking the above characterizations, it can be recognized that the prepared Al-Ni-P samples were in $(AlPO_4)m/Ni_2P$ composite structure, in which the molar ratios m of two components were in the range of 0.05–0.2; the reference Ni-P and Al-P samples were mainly present as $Ni_2P$ and $AlPO_4$.

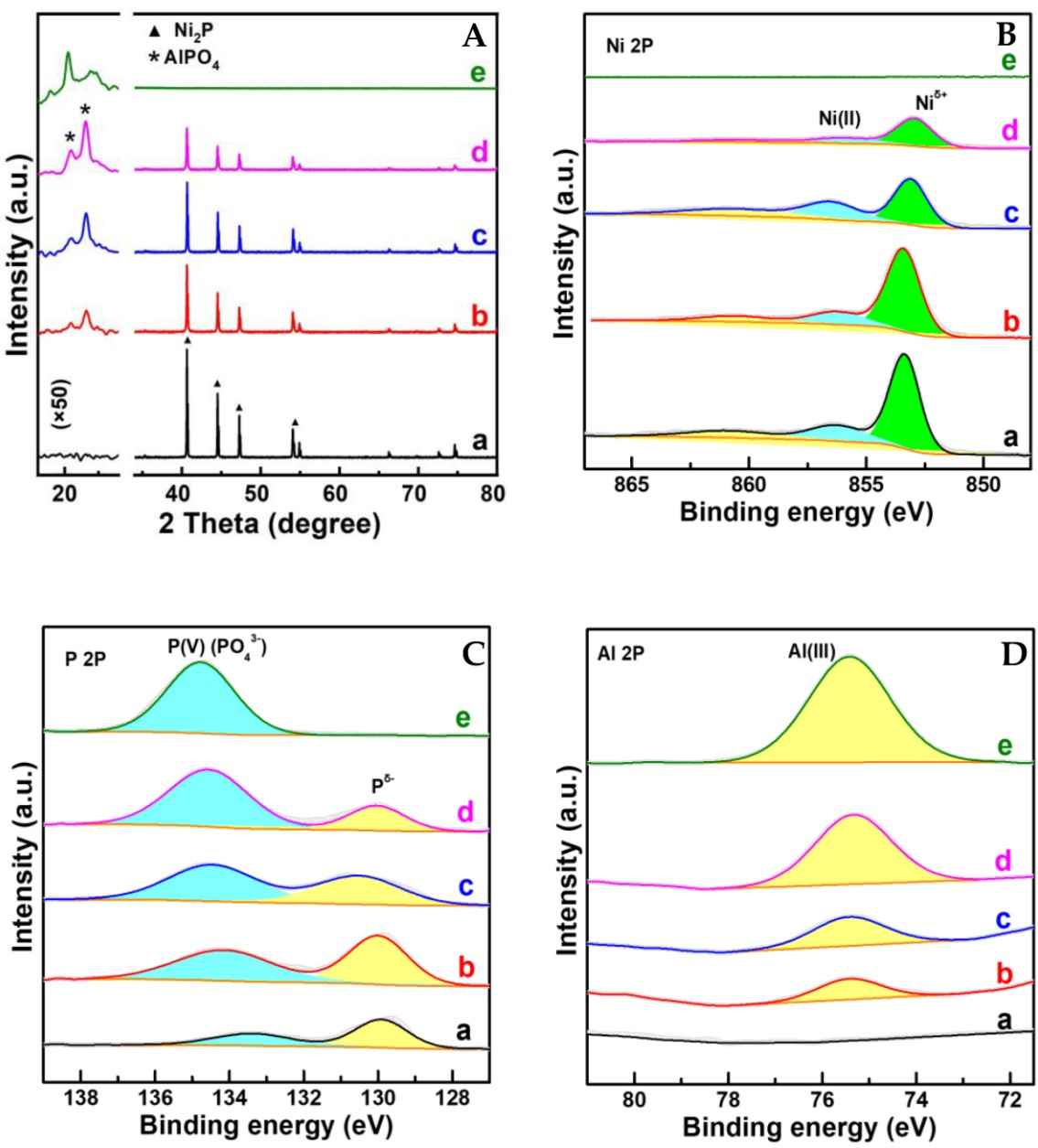

**Figure 1.** (**A–D**) XRD patterns and XPS spectra of samples. a. $Ni_2P$; b. $(AlPO_4)_{0.05}/Ni_2P$; c. $(AlPO_4)_{0.1}/Ni_2P$; d. $(AlPO_4)_{0.2}/Ni_2P$; e. $AlPO_4$. Note: The intensity of pattern at 2 theta around 20 degree in (**A**) was enlarged as 50 times as that of original pattern.

**Table 1.** Surface physical chemistry parameters of typical catalysts and their performances.

| Catalysts | Surface Composition (Al:Ni) | $N_{HA}$ ($\mu mol/g_{cat}$) | $N_{Acid}$ ($\mu mol/g_{cat}$) | Cinnamyl N-Propyl Ether Yield (%) | Cinnamyl Isopropyl Ether Yield (%) |
|---|---|---|---|---|---|
| $AlPO_4$ | a 1.2 | 11.6 | 1164.1 | - | - |
| $(AlPO_4)_{0.2}/Ni_2P$ | 2.3 | - | - | 61.9 | 59.2 |
| $(AlPO_4)_{0.1}/Ni_2P$ | 0.9 | 16.6 | 729.7 | 97.1 | 77.4 |
| $(AlPO_4)_{0.05}/Ni_2P$ | 0.3 | - | - | 64.1 | 64.0 |
| $Ni_2P$ | - | 0 | 244.0 | 14.6 | 24.2 |

-: none or it not being possible to determine the value; a Al:P; surface composition (Al:Ni): Surface composition as Al:Ni atomic ratio determined through XPS; $N_{HA}$: the moles of the site calculated from the H desorption amount; $N_{Acid}$: number of acid site calculated from $NH_3$-TPD profile.

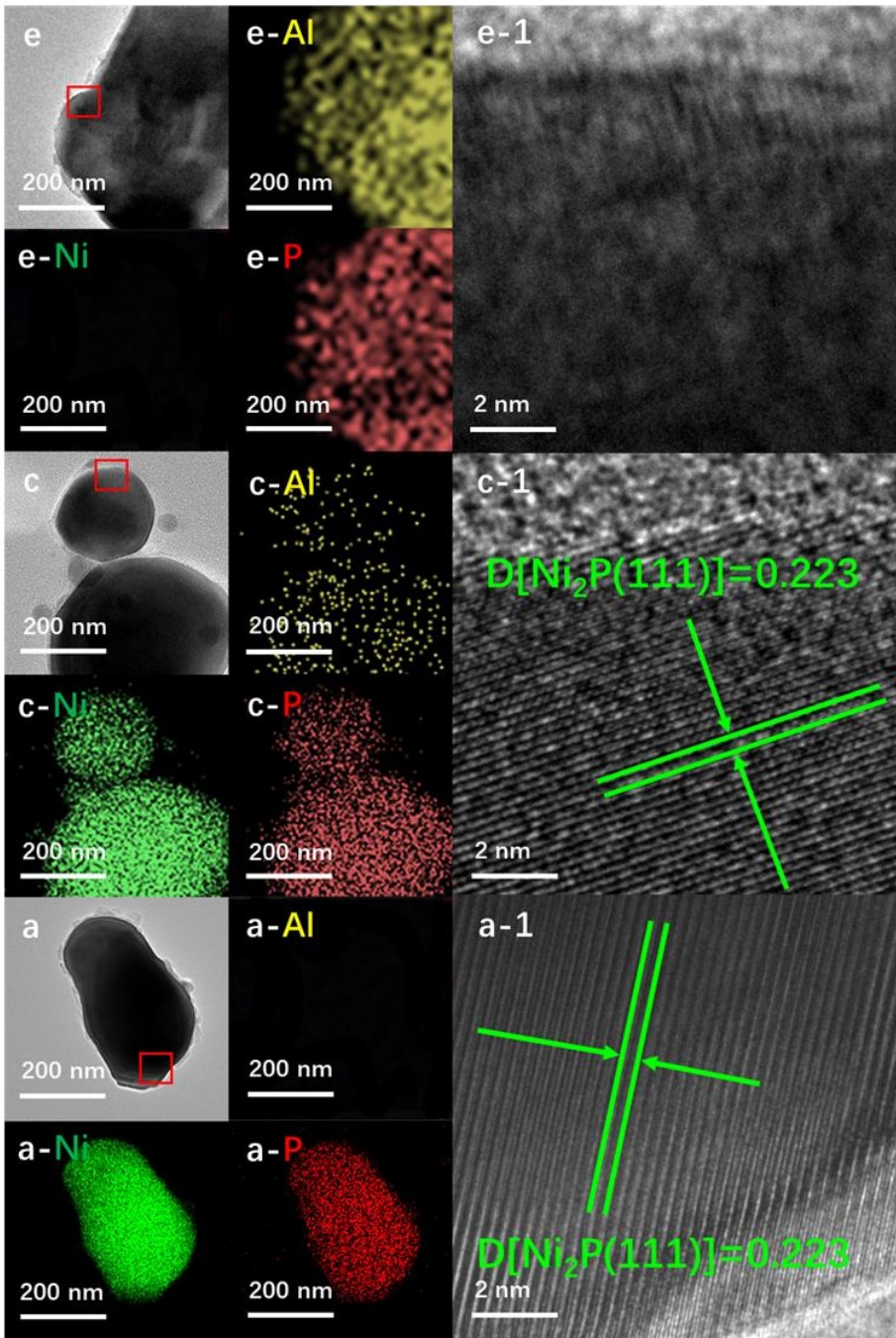

**Figure 2.** EDS mapping and HRTEM (enlarged from the marked area of the first column) images for the representative samples. a. $Ni_2P$; c. $(AlPO_4)_{0.1}/Ni_2P$; e. $AlPO_4$.

## 2.2. Catalytic Performance of System

The objective of this work is to investigate the potential of prepared Al-Ni-P composite catalysts for directly obtaining unsaturated ether (UE) from the hydrogenation transformation of unsaturated aldehyde (UA) reactants in alcohol solution. A full screen of the reaction network for the transformation as given in Scheme 1 with citral as the example substrate. It was shown that the transformation into the desired UE product was obstructed by the partial hydrogenation of two double bonds of UA, so the selectivity or yield to UE was the crucial aspect to estimate the efficiency of the catalysis systems.

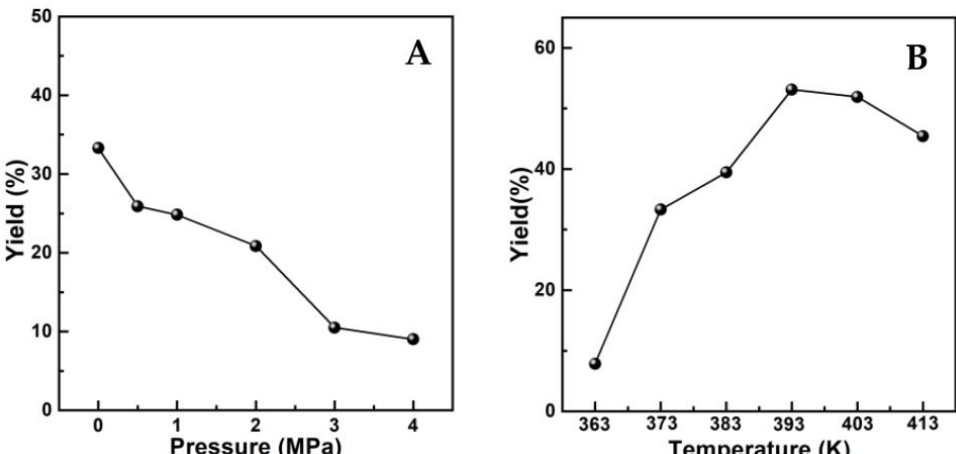

**Scheme 1.** The hydrogenation transformation network with citral as typical unsaturated aldehyde in alcohol solution.

According to reports [11,12], the selectivity or yield to UE was mainly impacted by two factors. One was reaction conditions such as $H_2$ pressure and temperature, the other was the nature of alcohol. The influences of $H_2$ pressure and temperature on the catalytic performance of current systems were shown by the variations in UE yield with altered reaction conditions in Figure 3, where $(AlPO_4)_{0.2}/Ni_2P$ was employed as an example for the hydrogenation transformation of cinnamaldehyde (CA) into cinnamyl isopropyl ether (CIPE) in isopropanol. It was shown that the yield of the desired CIPE product would be greatly cut down by increasing $H_2$ pressure from 0.1 MPa to 4.0 MPa, since high $H_2$ pressure would obviously increase the speed of the partial hydrogenation of CA itself to impede the transformation into CIPE. This observation is in agreement with the literature [11,12]; thus, pressure as low as 0.1 MPa was determined as the most favorable $H_2$ feedstock condition for the transformation in this work. Under such low $H_2$ pressure, temperature beyond 373 K was testified to be crucial for obtaining CIPE in a meaningful yield (>30%), and the highest yield could be achieved around 393 K, so 393 K was employed in the following catalytic measurements.

**Figure 3.** (**A**) Effect of $H_2$ pressure on UE yield. Reaction conditions: 1 mmol CA, 50 mg catalyst, 120 min, 373 K; (**B**) Influence of reaction temperature on UE yield. Reaction conditions: 1 mmol CA, 50 mg catalyst, 120 min, 0.1 MPa $H_2$.

Aside from the reaction conditions, the structure of alcohol also played an important role in altering reaction performance. In general, the relatively high yield (up to 97%) of UE could be achieved in primary alcohol; however, when secondary alcohol was employed, the yield of UE would be quenched around 30% [11,12]. For clarifying the corresponding feature on $(AlPO_4)m/Ni_2P$ catalysts, the catalytic performance of prepared catalysts were testified using the hydrogenation of cinnamaldehyde (CA) in propanol and isopropanol,

respectively under 0.1 MPa $H_2$ and 393 K. The dependence of CA conversion and UE yield on the composition of the catalyst is given in Figure 4A,B. In comparison with low conversion (around 40%) and yield ($\leq$22%) on $Ni_2P$, both CA conversion and UE yield are greatly enhanced on $(AlPO_4)m/Ni_2P$, although $AlPO_4$ itself is not active for the reaction; the highest yield of cinnamyl n-propyl ether (CNPE), up to 97% from hydrogenation coupling CA with propanol, was achieved on $(AlPO_4)_{0.1}/Ni_2P$. When coupling CA with isopropanol, cinnamyl isopropyl ether (CIPE) could also be efficiently harvested in a yield of 77% on $(AlPO_4)_{0.1}/Ni_2P$. To investigate the reuse potential of the catalysts, cycle tests by separating the used catalysts from the previous reaction solution and transferring them into fresh reaction solution were performed in succession. It was demonstrated that the notable yield of CIPE (around 60%) could remain constant (Figure 4C) in six cycles even on $(AlPO_4)_{0.05}/Ni_2P$, with the lowest addition of $AlPO_4$ among $(AlPO_4)m/Ni_2P$ samples; after six cycles, a gradual decrease in performance was exhibited, which could be attributed to the obvious accumulation of catalyst loss in mass from the separation process during cycle tests. This understanding was identified by the following observations: (1) the insignificant metal leakage remained during continuous catalytic cycles and (2) the almost unchanged surface state of the cycle-used sample compared with its fresh counterpart (Ni 2p XPS spectra in Figure 4D). These results illuminated that $AlPO_4/Ni_2P$ composite could act as an efficient and robust catalyst for directly acquiring UE from the hydrogenation transformation of UA in alcohol.

The catalytic performance of the current $(AlPO_4)m/Ni_2P$ catalyst is further compared with that of reported catalysts in Table 2. Under mild reaction conditions, $(AlPO_4)_{0.1}/Ni_2P$ offered a CNPE yield as high as 97%, which was comparable with precious metal catalysts; surprisingly, when coupling CA with isopropanol, the attractive yield of cinnamyl isopropyl ether (CIPE) of up to 77% could be also obtained on $(AlPO_4)_{0.1}/Ni_2P$ at 393 K with 0.1 MPa $H_2$, which was six-fold higher than the yield of CIPE over a reported Pt catalyst under 3.0 MPa $H_2$ and 408 K [12]. The comparison suggested that $(AlPO_4)m/Ni_2P$ could be a more competitive candidate catalyst than reported catalysts for the transformation in view of its comprehensive advantages of low cost, high efficiency and convenient operation.

### 2.3. Catalytic Origin of AlPO_4/Ni_2P Composites

To understand the transformation behavior of the reaction, in situ IR measurement was performed by monitoring the spectra change after the introduction of $H_2$ on $(AlPO_4)_{0.2}/Ni_2P$ with CA and propanol chemisorbed previously, as shown in Figure 5A. The chemisorbed CA on the sample was reflected by the feature vibration peaks for a C=O bond in the wavenumber range of 1650–1700 $cm^{-1}$ and a C=C bond in 1450–1500 $cm^{-1}$ [19,20] present at t = 0 min; then, C=O vibration was gradually decreased with the introduction of $H_2$, while the vibration belonging to a C-O-C bond that centered around 1100 $cm^{-1}$ [21,22] occurred and increased with the prolonged time. Notably, the intensity of the C=C bond in the whole process was almost not influenced, indicting the C=C bond was preserved during the reaction. In addition, by employing $D_2$ to replace $H_2$, the m/z for the UE product shifted positively with two units in mass spectra (in Figure 5B), indicating that the reaction process involves the hydrogenation step. These results clearly illustrate that the UE product on $AlPO_4/Ni_2P$ composite catalysts was formed from a selective hydrogenation coupling process between carbonyl and alcohol. In view of this, the whole process could be figured out as a hydrogenation dehydration reaction between UA and alcohol, which was also supported by the insight of the literature [11,12]. The possible transformation steps are given in Scheme 2. The reaction route implied that both the hydrogen activation site and dehydration site were required for the reaction. Here, $H_2$-TPD and $NH_3$-TPD were employed to investigate the properties of the active site of the prepared catalysts, as shown in Figure 5C,D. In contrast to the insignificant signal on $Ni_2P$, the obvious H-desorption peaks were observed on $AlPO_4$ and $(AlPO_4)_{0.1}/Ni_2P$, indicating that the $AlPO_4$ component could act as a hydrogen activation site on $AlPO_4/Ni_2P$ composite catalysts; furthermore, the more distinct $NH_3$-desorption peaks occurred on $AlPO_4$ and $(AlPO_4)_{0.1}/Ni_2P$ in comparison with $Ni_2P$, and the calculated acid site number (listed in Table 1) confirmed

that integration of $AlPO_4$ on $Ni_2P$ would offer more acid sites for composite catalysts, which could act as the active site for the dehydration process [23,24]. Therefore, the catalytic feature for the reaction on $AlPO_4/Ni_2P$ composite catalysts could be attributed to the bi-functional interface (hydrogen activation and acid catalysis) formed by the integration of $AlPO_4$ on $Ni_2P$. As far as the role of the $Ni_2P$ component is concerned, it has been testified that the $Ni_2P$ surface was responsible for the chemical adsorption or activation of UA [25], so a proper ratio between $AlPO_4$ and $Ni_2P$ could be another factor to govern the catalytic performance of $(AlPO_4)m/Ni_2P$ catalysts. According to catalytic measurements, m was determined as 0.1–0.2 for acquiring UE in the highest yield in this work.

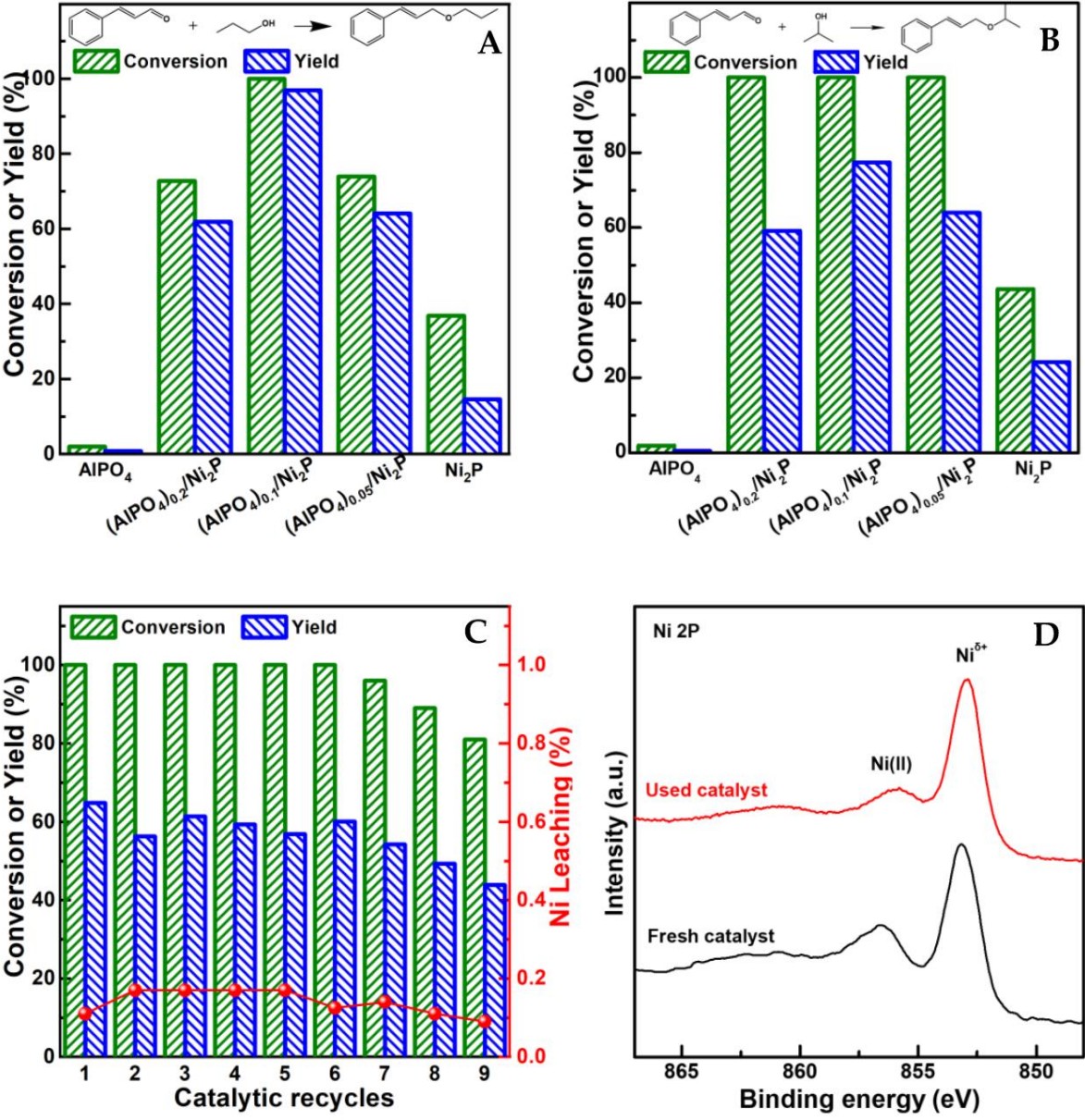

**Figure 4.** Typical catalytic performance of prepared catalysts. (**A**,**B**) Comparison of conversion of cinnamaldehyde (CA) and yield of unsaturated ether for hydrogenation transformation of CA in propyl alcohol among catalysts. (**C**) Catalytic performance variations of reused $(AlPO_4)_{0.05}/Ni_2P$ for hydrogenation transformation of CA in isopropyl alcohol in catalytic recycles. (**D**) Comparison of surface properties with Ni 2p XPS spectra between fresh and cycle-used $(AlPO_4)_{0.05}/Ni_2P$. Reaction conditions: CA (1 mmol), catalyst (50 mg), dodecane (100 μL), reaction solvent ((**A**) n-propyl alcohol, (**B**) isopropyl alcohol, 15 mL), 0.1 MPa $H_2$, 393 K, reaction period ((**A**) 2 h, (**B**) 4 h).

**Table 2.** Comparison of performance of different catalytic systems for hydrogenation transformation of cinnamaldehyde to cinnamyl alkyl ether in alcohol.

| Catalysts | Reaction Conditions | Solvent | Conv./% | Sel./% | Ref. |
|---|---|---|---|---|---|
| Au/TiO$_2$ | 333 K, 0.1 MPa, 5 h | ethanol | 50 | 33 | [11] |
| Pt@GS | 408 K, 3 MPa, 3 h | n-propanol | 99 | 97 | [12] |
| | | isopropanol | 37 | 33 | |
| (AlPO$_4$)$_{0.1}$/Ni$_2$P | 393 K, 0.1 MPa, 2 h | n-propanol | 99 | 97.1 | This work |
| | 393 K, 0.1 MPa, 2 h | isopropanol | 99 | 77.4 | |

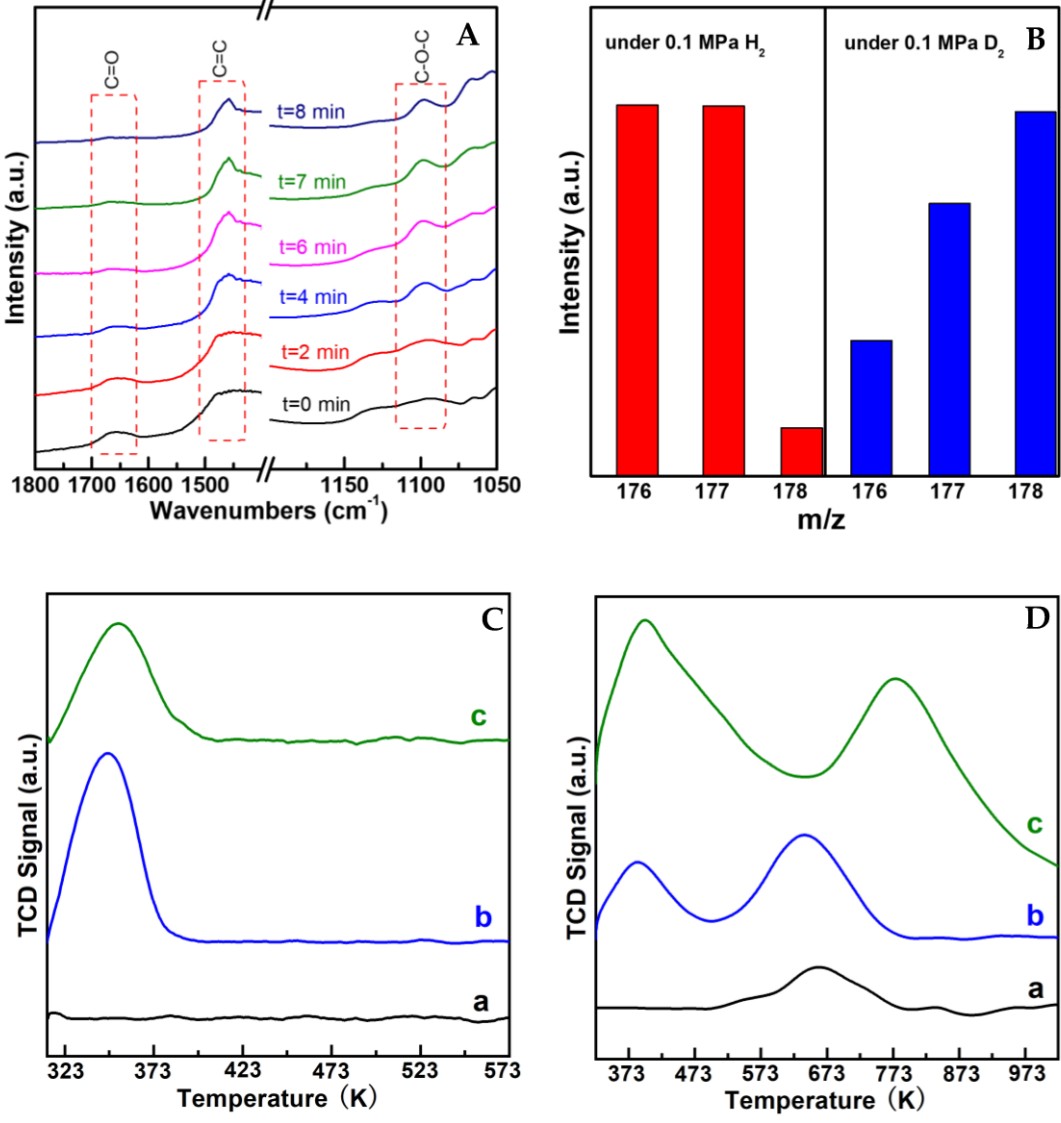

**Figure 5.** Mechanism investigation results over typical catalysts. (**A**) In situ FT-IR spectra by introducing H$_2$ on (AlPO$_4$)$_{0.2}$/Ni$_2$P with cinnamaldehyde(CA) and propanol chemisorbed previously. (**B**) Comparison of product distribution from mass spectra by replacing H$_2$ with D$_2$. Reaction conditions: CA (1 mmol), catalyst (50 mg), dodecane (100 µL), n-propyl alcohol (15 mL), 393 K, 4 h, 0.1 MPa H$_2$ or D$_2$. (**C,D**) H$_2$-TPD and NH$_3$-TPD profiles over representative catalysts. a. Ni$_2$P; b. (AlPO$_4$)$_{0.1}$/Ni$_2$P; c. AlPO$_4$.

**Scheme 2.** Possible reaction pathway for hydrogenation transformation of unsaturated aldehyde (UA) to unsaturated ether (UE) in alcohol.

### 2.4. Application Potential of Present Catalytic Systems

A scope of C1–C5 alcohol was further employed as the reaction solvent for the hydrogenation of CA over $AlPO_4/Ni_2P$ catalysts. In addition, citral was also used as a substrate to further investigate the applicability of the system. Typical results with CA and citral as reactants are collected in Tables 3 and 4, respectively. The GC-MS analysis spectra for every reaction are given in Figures S1–S18 in the supporting information. For comparison, n-hexane and water were also employed as the solvent. Only the C=C hydrogenation product (HCAL) or saturated product (HCOL) from the hydrogenation of CA were found in n-hexane and water; in contrast, UE was present as the main product with a yield of 51–97% for CA hydrogenation in C1–C5 alcohol, no matter whether the solvent was primary or secondary alcohol, except for C5 secondary alcohol (2-Pen-OH). For citral, an attractive UE yield of 70–93% was obtained in primary C1-C5 alcohol, while in secondary alcohols, a considerable UE yield of 11–40% could be also achieved under 0.1 MPa $H_2$ and 393 K. These results further confirmed that high-grade unsaturated ether could be directly and feasibly acquired from selective hydrogenation coupling unsaturated aldehyde with alcohol with attractive efficiency upon bi-functional $AlPO_4/Ni_2P$ catalysts.

**Table 3.** Catalytic hydrogenation performance over $(AlPO_4)_{0.1}/Ni_2P$ catalyst for cinnamic aldehyde with controlled reactants and reaction conditions. [a]: Diacetal; [b]: Performed under $N_2$ pressure (without $H_2$); HCAL: Phenylpropionaldehyde; HCOL: Phenylpropanol; UE: Unsaturated ether; Reaction conditions: 1 mmol of cinnamic aldehyde, 100 μL of dodecane, 50 mg of $(AlPO_4)_{0.1}/Ni_2P$ catalyst in 15 mL of solvent.

| Entry | Reactant | Reaction Solvent | Conditions | Con. (%) | Sel. (%) | | | UE Yield (%) |
|---|---|---|---|---|---|---|---|---|
| | | | | | HCAL | HCOL | UE | |
| 1 | | n-hexane | 0.1 MPa, 393 K, 2 h | 21.2 | 99 | - | - | - |
| 2 | | water | 0.1 MPa, 393 K, 2 h | >99 | 99 | - | - | - |
| 3 | | | 2.0 MPa, 393 K, 2 h | >99 | - | 99 | - | - |
| 4 | | Me-OH | 0.1 MPa, 393 K, 2 h | >99 | - | - | 98.3 [a] | - |
| 5 | | Et-OH | 0.1 MPa, 393 K, 2 h | >99 | 2.3 | - | 97.7 | 97.7 |
| 6 | Cinnamic | n-Pr-OH | 0.1 MPa, 393 K, 2 h | >99 | 2.9 | - | 97.1 | 97.1 |
| 7 | aldehyde | i-Pr-OH | 0.1 MPa, 393 K, 4 h | >99 | 20.8 | 1.8 | 77.4 | 77.4 |
| 8 | | n-Bu-OH | 0.1 MPa, 393 K, 4 h | >99 | 3.3 | - | 96.7 | 96.7 |
| 9 | | s-Bu-OH | 0.1 MPa, 393 K, 4 h | >99 | 44.3 | 4.5 | 51.2 | 51.2 |
| 10 | | n-Am-OH | 0.1 MPa, 393 K, 4 h | >99 | 18.4 | 1.6 | 80.0 | 80.0 |
| 11 | | 2-Am-OH | 0.1 MPa, 393 K, 4 h | 63.2 | 85.5 | - | 14.5 | 9.2 |
| 12 [b] | | n-Pr-OH | 0.1 MPa, 393 K, 2 h | - | - | - | - | - |

**Table 4.** Catalytic hydrogenation performance over $(AlPO_4)_{0.1}/Ni_2P$ catalyst for citral with varying reactants and reaction conditions. HCAL: Citronellol; HCOL: 3,7-Dimethyloctanol; UE: Unsaturated ether; Reaction conditions: 1 mmol of citral, 100 μL of dodecane, 50 mg of $(AlPO_4)_{0.1}/Ni_2P$ catalyst in 15 mL of solvent.

| Entry | Reactant | Reaction Solvent | Conditions | Con. (%) | Sel. (%) | | | UE Yield (%) |
|---|---|---|---|---|---|---|---|---|
| | | | | | HCAL | HCOL | UE | |
| 1 | | Me-OH | 0.1 MPa, 393 K, 2 h | >99 | 4.7 | 2.2 | 93.2 | 93.2 |
| 2 | | Et-OH | 0.1 MPa, 393 K, 2 h | >99 | 12.9 | 4.2 | 83.0 | 83.0 |
| 3 | | n-Pr-OH | 0.1 MPa, 393 K, 2 h | >99 | 18.5 | 2.6 | 78.9 | 78.9 |
| 4 | Citral | i-Pr-OH | 0.1 MPa, 393 K, 4 h | >99 | 57.4 | 2.7 | 39.8 | 39.8 |
| 5 | | n-Bu-OH | 0.1 MPa, 393 K, 4 h | >99 | 21.9 | 5.2 | 72.9 | 72.9 |
| 6 | | s-Bu-OH | 0.1 MPa, 393 K, 4 h | >99 | 73.8 | 7.7 | 18.5 | 18.5 |
| 7 | | n-Am-OH | 0.1 MPa, 393 K, 4 h | >99 | 26.1 | 3.6 | 70.3 | 70.3 |
| 8 | | 2-Am-OH | 0.1 MPa, 393 K, 4 h | 68.1 | 87.3 | - | 12.7 | 11.1 |

　　　　Based on the above measurements, some details could be further elucidated. **(1)** The product distribution. The hydrogenation transformation of UA in alcohol was a complicated reaction network, in which reductive etherification was accompanied by the competitive hydrogenation of UA itself, and the direct hydrogenation products of UA such as HCAL and HCOL were present as the main by-products. Firstly, the selectivity to the desired UE product was mainly governed by $H_2$ pressure and the structure of alcohol: high $H_2$ pressure (Figure 3A) would greatly increase the speed of the hydrogenation of UA to HCAL and HCOL; secondary alcohol would clearly suppress the reductive etherification between UA and alcohol due to the branch barrier effect in comparison with primary alcohol (Tables 3 and 4); and both of these two factors would lead to the low yield of the UE product. In light of this, one may further question the existence of saturated ether product. In our case, such products were not observed. The first reason could be the low pressure of $H_2$ (0.1 MPa); as illuminated, the hydrogenation of the UA substrate itself in primary alcohol could be confined to an insignificant level, suggesting the direct hydrogenation of the C=C bond in the UA substrate was greatly decelerated under such a low pressure of $H_2$ feedstock, so production of saturated ether from further hydrogenation of UE could also be inhibited in this condition. In addition, another source of saturated ether product formed by the direct dehydration process of alcohol was also not evident in the current reaction network, which could be attributed to the absence of a proton donor in the current heterogeneous catalytic system, since a dissociative proton or proton donor was necessary for facilitating direct dehydration between two alcohol molecules [26,27]. **(2)** The correlation between greenness and efficiency of current catalytic system. Although the current $AlPO_4/Ni_2P$ catalyst showed attractive performance for the reaction among reported catalytic systems (Table 2), we must admit that the yield of UE, especially in secondary alcohol, was far below the ideal level. As mentioned, the low yield of UE could be mainly attributed to the branch barrier effect of secondary alcohol. A similar phenomenon was always an inevitable challenge in fine chemical synthesis no matter what catalyst was employed. Moreover, the main by-products such as HCAL and HCOL present in the current system were also important and useful compounds produced from the UA substrate, and at least these compounds cannot be classified as waste or toxicant. In addition, the reaction network was more environmentally friendly than other possible reaction routes involving hazardous additives [7–10], so it can be recognized that these features gave the catalytic system potential to serve as an efficient and green system for feasibly acquiring high-grade unsaturated ether.

## 3. Materials and Methods

### 3.1. Materials

All chemical reagents in the chemical analysis were commercially obtained and used directly without further purification. Nickel (II) nitrate ($Ni(NO_3)_3 \cdot 6H_2O$), diammonium hydrogen phosphate (($NH_4)_2HPO_4$) and aluminum hydroxide ($Al(OH)_3$) were purchased from Sinopharm Chemical Reagent Co., Ltd. (Shanghai, China) Cinnamic aldehyde (CA), and citral and dodecane were obtained from Innochem Reagent Co., Ltd. (Beijing, China).

### 3.2. Preparation of Catalysts

Samples with different Al/Ni molar ratios were prepared in a one-pot hydrothermal reaction system. The required $Al(OH)_3$ and $Ni(NO_3)_2 \cdot 6H_2O$ at different molar ratios were dissolved in deionized water, and then $(NH_4)_2HPO_4$ aqueous solution was dropped into the solution. After the solution was mixed at a stirring speed of 450 rpm for 1 h, it was transferred into a hydrothermal kettle and reacted at 453 K for 24 h. Then, the reactor was naturally cooled to room temperature, and the obtained precipitate was centrifuged out and washed with deionized water for several times, then dried at 343 K for 12 h to obtain the precursor. The precursors were calcined in air at 873 K for 4 h, and then reduced for 2 h in a hydrogen flow with a rate of 50 mL/min to acquire catalyst samples.

### 3.3. Characterizations

The actual compositions of metal components (Al and Ni) in catalyst samples were analyzed on an Agilent Technologies 5100 (Santa Clara, CA, USA) inductively coupled plasma optical emission spectrometer (ICP-OES).

The X-ray diffraction (XRD) data of the sample powders were collected on a Puxi DX-3 diffractometer (Manchester, UK) operated at 40 kV, 40 mA for Cu K$\alpha$ ($\lambda$ = 1.5406 Å), with a scan speed of 8°/min at 2θ range of 10°–80°.

The X-ray photoelectron spectroscopy (XPS) of the samples was measured using an Axis Ultra DLD device (Manchester, UK) with monochromated Al K$\alpha$ (1486.6 eV) as the excitation source. The C 1s peak at 284.8 eV was used to calibrate the binding energies for the samples.

The transmission electron microscopy (TEM) images of test samples were taken with a JEOL JEM-2100 (Akishima City, Tokyo, Japan) microscope operating at 200 kV.

The temperature-programmed desorption of $H_2$ [$H_2$-TPD] or $NH_3$ [$NH_3$-TPD] of the catalyst was performed on a Micromeritics-Auto-Chem II 2920 (Norcross, GA, USA) chemisorption analyzer. Before the tests, 50 mg of sample was sealed in the reactor, pretreated with an Ar flow (30 mL min$^{-1}$, 30 min, 573 K), and the system was cooled to room temperature under protection of Ar. For $H_2$-TPD measurements, the system was saturated by a flow of $H_2$ (30 mL min$^{-1}$) for 60 min, and then by removing the free adsorbate with an argon stream at the rate of 30 mL min$^{-1}$ for 30 min. Subsequently, the sample was heated to 573 K at a heating rate of 10 K min$^{-1}$, and desorption signals were monitored using TCD. For $NH_3$-TPD measurements, after adsorbing $NH_3$ at a flow rate of 30 mL min$^{-1}$ for 1 h, the sample was blown with pure He for 1 h to remove physically adsorbed molecules, then heated from room temperature at a ramp rate of 10 K min$^{-1}$, and the desorption signal was continuously monitored using TCD.

The in situ FT-IR spectra of the catalysts were performed on an in situ diffuse reflectance infrared spectrometer with a resolution of 4 cm$^{-1}$ and a scanning number of 32 composed of a Nicolet IS10 (Madison, WI, USA) infrared spectrometer (Thermo Fisher, Waltham, MA, USA) and a heat chamber for diffuse IRTM accessory (Pike Technologies), and the infrared spectral signals were collected across a ZnSe disk. In a typical procedure, the catalyst fixed in the in situ chamber was firstly blown by Ar at 573 K for 30 min to obtain a relatively clean surface, then the system was cooled down to the reaction temperature 393 K and the reactants (CA) from the n-propanol solution were bubbled into the chamber by Ar (30 mL min$^{-1}$) for 30 min, then the adsorption infrared signals were collected from cycled scanning between 600–4000 cm$^{-1}$ under the constant temperature of 393 K until the

stable signals were obtained, and $H_2$ flow at 30 mL min$^{-1}$ was introduced in the chamber to collect infrared signals as in situ data.

### 3.4. Catalytic Measurements

Without specification, the reductive etherification of unsaturated aldehydes (CA and citral) were all performed at 393 K under 0.1 MPa $H_2$ in a 100 mL Teflon reactor with a stainless-steel autoclave heater equipped with liquid-sampling device. Typically, the reaction mixture was composed of 1 mmol CA or citral, 50 mg catalyst and 15 mL alcohol following the similar reaction composition used in the literature [11,12]; prior to the reaction, the reaction system was purged 5 times with $H_2$ to exclude air by stirring at a speed of 350 rpm. After the reaction was performed in a certain period (2–4 h), the reaction mixture was cooled to ambient temperature and the catalyst was filtered off, then the reaction mixture was used for quantitative analysis. Due to the limited amount of objective compounds, we failed to obtain credible data from chromatographed measurements and to isolate enough pure compound from the mixture for NMR measurements, which could also be the reason for the absence of these analysis measurements in similar research [11,12,28,29]. Following the qualitative method employed by these works in the literature, the composition of the reaction mixture was analyzed on a gas chromatography mass spectrometer (GC-MS 7890B-5977A equipped with an Agilent 19091S-433 (Santa Clara, CA, USA) capillary column (HP-5MS, 30 m × 250 μm × 0.25 μm)); the conversion of unsaturated aldehydes (UA) was calibrated according to previously determined composition–standard curves. The yields of products (UE, HCAL and HCOL) were comprehensively determined by the composition–standard curves and internal standard index reagent (dodecane).

### 4. Conclusions

In summary, a series of Al-Ni-P catalysts in $AlPO_4/Ni_2P$ composite structure were innovated and testified as competitive candidate catalysts for the direct and efficient acquirement of unsaturated ether (UE) from the selective hydrogenation coupling of unsaturated aldehyde with alcohol under 393 K and 0.1 MPa $H_2$. Aside from $H_2$ pressure, the structure of alcohol was testified to be the key factor to govern the yield of UE; by properly manipulating the catalyst composition, a UE yield as high as 70–97% from primary alcohol or in the range of 10–77% from secondary alcohol was achieved on $AlPO_4/Ni_2P$ samples. The unique bi-function of hydrogen activation and acid catalysis along the $AlPO_4/Ni_2P$ interface was demonstrated to be responsible for the attractive performance. The features of low cost, attractive performance and environmental friendliness offer the potential of the system to directly produce high-grade unsaturated ether in related advanced synthesis of fine chemicals.

**Supplementary Materials:** The following supporting information can be downloaded at: https://www.mdpi.com/article/10.3390/catal13020439/s1, Figure S1: (a) GC conversion of the phenylpropanal from the reaction of 1 mmol of cinnamaldehyde and n-hexane at 120 °C under 0.1 MPa H2 pressure and GC-MS spectra and its fragmentation pattern of (b) Cinnamaldehyde (starting compound); (c) Dodecane (internal standard); (d) Phenylpropanal. Figure S2: (a) GC conversion of the phenylpropanol from the reaction of 1 mmol of cinnamaldehyde and water at 120 °C under 2 MPa H2 pressure; (b) GC-MS spectra and its fragmentation pattern of phenylpropanol. Figure S3: (a) GC conversion of the (4,4-dimethoxybutyl) benzene from the reaction of 1 mmol of cinnamaldehyde and methanol at 120 °C under 0.1 MPa H2 pressure; (b) GC-MS spectra and its fragmentation pattern of (4,4-dimethoxybutyl) benzene. Figure S4: (a) GC conversion of the unsaturated ether from the reaction of 1 mmol of cinnamaldehyde and ethanol at 120 °C under 0.1 MPa H2 pressure; (b) GC-MS spectra and its fragmentation pattern of unsaturated ether. Figure S5: (a) GC conversion of the unsaturated ether from the reaction of 1 mmol of cinnamaldehyde and n-propanol at 120 °C under 0.1 MPa H2 pressure; (b) GC-MS spectra and its fragmentation pattern of unsaturated ether. Figure S6: (a) GC conversion of the unsaturated ether from the reaction of 1 mmol of cinnamaldehyde and isopropyl alcohol at 120 °C under 0.1 MPa H2 pressure; (b) GC-MS spectra and its fragmentation pattern of unsaturated ether. Figure S7: (a) GC conversion of the unsaturated ether from the reaction of 1 mmol

of cinnamaldehyde and n-butanol at 120 °C under 0.1 MPa H2 pressure; (b) GC-MS spectra and its fragmentation pattern of unsaturated ether. Figure S8: (a) GC conversion of the unsaturated ether from the reaction of 1 mmol of cinnamaldehyde and sec-butanol at 120 °C under 0.1 MPa H2 pressure; (b) GC-MS spectra and its fragmentation pattern of unsaturated ether. Figure S9: (a) GC conversion of the unsaturated ether from the reaction of 1 mmol of cinnamaldehyde and n-pentanol at 120 °C under 0.1 MPa H2 pressure; (b) GC-MS spectra and its fragmentation pattern of unsaturated ether. Figure S10: (a) GC conversion of the unsaturated ether from the reaction of 1 mmol of cinnamaldehyde and 2-pentanol at 120 °C under 0.1 MPa H2 pressure; (b) GC-MS spectra and its fragmentation pattern of unsaturated ether. Figure S11: (a) GC conversion of the unsaturated ether from the reaction of 1 mmol of citral and methanol at 120 °C under 0.1 MPa H2 pressure; (b) GC-MS spectra and its fragmentation pattern of unsaturated ether. Figure S12: (a) GC conversion of the unsaturated ether from the reaction of 1 mmol of citral and ethanol at 120 °C under 0.1 MPa H2 pressure; (b) GC-MS spectra and its fragmentation pattern of unsaturated ether. Figure S13: (a) GC conversion of the unsaturated ether from the reaction of 1 mmol of citral and n-propanol at 120 °C under 0.1 MPa H2 pressure; (b) GC-MS spectra and its fragmentation pattern of unsaturated ether. Figure S14: (a) GC conversion of the unsaturated ether from the reaction of 1 mmol of citral and isopropyl alcohol at 120 °C under 0.1 MPa H2 pressure; (b) GC-MS spectra and its fragmentation pattern of unsaturated ether. Figure S15: (a) GC conversion of the unsaturated ether from the reaction of 1 mmol of citral and n-butanol at 120 °C under 0.1 MPa H2 pressure; (b) GC-MS spectra and its fragmentation pattern of unsaturated ether. Figure S16: (a) GC conversion of the unsaturated ether from the reaction of 1 mmol of citral and sec-butanol at 120 °C under 0.1 MPa H2 pressure; (b) GC-MS spectra and its fragmentation pattern of unsaturated ether. Figure S17: (a) GC conversion of the unsaturated ether from the reaction of 1 mmol of citral and n-pentanol at 120 °C under 0.1 MPa H2 pressure; (b) GC-MS spectra and its fragmentation pattern of unsaturated ether. Figure S18: (a) GC conversion of the unsaturated ether from the reaction of 1 mmol of citral and 2-pentanol at 120 °C under 0.1 MPa H2 pressure; (b) GC-MS spectra and its fragmentation pattern of unsaturated ether.

**Author Contributions:** Y.X. and H.Z.: Validation, Formal analysis, Writing—Original Draft, Investigation, Terms, Conceptualization, Methodology. D.Z., S.W., S.D. and C.C.: Writing—Review and Editing, Visualization, Supervision, Project Administration, Resources. All authors have read and agreed to the published version of the manuscript.

**Funding:** This research was financially supported by the National Natural Science Foundation of China (NSFC, No. 21563018, No. 21663016 and No. 22062014) and the Key Laboratory of Nanchang City (Grant 2021NCZDSY-005).

**Data Availability Statement:** The data that support the findings of this study are available in the Supporting Information of this article.

**Conflicts of Interest:** The authors declare no conflict of interest.

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
