# Peer review of "Green and Efficient Acquirement of Unsaturated Ether from Direct and Selective Hydrogenation Coupling Unsaturated Aldehyde with Alcohol by Bi-Functional Al-Ni-P Heterogeneous Catalysts"

_catalysts, doi:10.3390/catal13020439_

Round 1
Reviewer 1 Report
This submitted paper describes the preparation of unsatured ethers (UE) resulting from a reaction carried out in heterogeneous catalysis between citral or cinnamaldehyde with various primary or secondary alcohols.
Analyzing this paper in detail, I notice some critical points.
The experimental section is lacking: it is not possible to understand what are precisely the operating conditions under which the various reactions take place. However, from the supporting information and tables it would appear that the reactions are conducted at 120°C. Yields are also reported, but how they are calculated is not explained. Since the NMRs of the UE obtained are not provided, it would seem that the authors report yields calculated at the GC and not of actually isolated products.
The used catalyst was not recovered and reused in further catalytic cycles. Why?
It would be appropriate to add a Scheme showing in dept the reaction mechanism.
Authors claim the high-efficiency of this protocol. Actually, from a synthetic point of view it is far from efficient!
Yields determined by GC and not of isolated products are most likely reported. Pure UEs have never been obtained: some by-products are always present in variable amounts. It is clear that an UE contaminated by by-products cannot be used, whatever the purpose. Then the reaction crudes should be chromatographed.
Assuming that the yields calculated at GC are similar to those of the isolated product, in many cases the yields are not satisfactory, invalidating the presumed efficiency of this synthetic protocol.
Finally, the authors claim the green aspects of this reaction; first of all it is difficult to consider a synthetic protocol, that involves heating to 120°C, green. Moreover, a green aspect would be the possibility of recovering and reusing the catalyst: but it seems that this is not done. The presence of the by-products considerably lowers the atom-economy of this reaction. From the data provided it is not possible to get an idea of ​​what the E-factor of this reaction could be.
In light of the above, I do not consider this paper suitable for publication
Reviewer 2 Report
The manuscript describes conversion of a,b-unsaturated aldehydes in the presence of hydrogen gas and alcohols to unsaturated ethers. In general, selective hydrogenation of a,b-unsaturated aldehydes is an important topic; and the conversion of them to unsaturated ether is another step forward. This topic is highly important.
Overall the authors have done a very nice job describing the results. Here are some remarks:
- my main concern is about Scheme 1. Going from citral to unsaturated ether one would need hydrogen as well. H2 should be added. Alternatively, the arrow should go from geraniol to unsaturated ether, in this case addition of alcohol and elimination of water would be correct.
- in Table 2, I am surprised that you do not form any saturated ether. I would like to ask authors to provide an explanation for that in the text, as I would expect to see some from at least coupling of the saturated alcohol, which is formed as the side product.
Once these 2 things are fulfilled, I am happy to suggest the article for the publication in this journal.
Reviewer 3 Report
The authors present interesting study on the possibility to obtain unsaturated ethers through the reductive etherification of unsaturated aldehydes and alcohols. This direction is not studied intensively because of the low yield of target product, thus. it is not used industrially. However, the authors developed novel catalysts which showed high effectiveness in the production of unsaturated ethers. The paper is written well and clearly. This work seems to be a completed study with the meaningfull results. I am in favor of the publication of this paper in the present form.
The main question addressed by the research is an increase in the yield of unsaturated ethers by the reductive etherification of unsaturated aldehydes and alcohols. This reaction is possible at all, but the main challenge is the low yield of the target product. In the presented work, authors designed a catalyst allowing the yield of unsaturated ethers to be increased sufficintly. The results presented are original and have high potential. As I mentioned, an increase in the yield of unsaturated ethers during direct synthesis from aldehydes and alcohols is a main gap in fine synthesis. So, the authors tried to solve this problem in their work. I beleive, an idea for the design of the catalyst which exhibit high activity and selectivity to unsaturated ethers is the main novelty of this work. In methodology, I suggest the authors to add any information about the calculation of the conversion and yield/selectivity. Also, an additional information about the quantitative analysis (use of standards, calybration etc.) can be added. I can recommend to expand the conclusions section with the numerical results. The references are appropriate. The presented tables and figures are clear and contain all the information needed.
Round 2
Reviewer 1 Report
I appreciate the efforts of the authors in revising the paper